# Firearm Deaths among Youth in the United States, 2007–2016

**DOI:** 10.3390/children10081359

**Published:** 2023-08-08

**Authors:** Theodore E. Trigylidas, Patricia G. Schnitzer, Heather K. Dykstra, Gia M. Badolato, Robert McCarter, Monika K. Goyal, Richard Lichenstein

**Affiliations:** 1Division of Emergency Medicine, Children’s National Hospital, 111 Michigan Ave NW, Washington, DC 20010, USA; gbadolat@childrensnational.org (G.M.B.); mgoyal@childrensnational.org (M.K.G.); 2Department of Pediatrics, The George Washington University School of Medicine and Health Sciences, 2300 I Street NW, Washington, DC 20037, USA; rjm.res@gmail.com; 3The National Center for Fatality Review & Prevention, 2395 Jolly Road Suite 120, Okemos, MI 48864, USA; pschnitzer@outlook.com (P.G.S.); hdykstra@mphi.org (H.K.D.); 4Department of Pediatrics, University of Maryland School of Medicine, 22 S. Greene Street, Baltimore, MD 21201, USA; rlichenstein@som.umaryland.edu

**Keywords:** firearm storage, child death review, firearm suicide, firearm homicide

## Abstract

We sought to compare risk factors contributing to unintentional, homicide, and suicide firearm deaths in children. We conducted a retrospective review of the National Fatality Review Case Reporting System. We included all firearm deaths among children aged 0–18 years occurring from 2007 to 2016. Descriptive analyses were performed on demographic, psychosocial, and firearm characteristics and their relationship to unintentional, homicide, and suicide deaths. Regression analyses were used to compare factors contributing to unintentional vs. intentional deaths. There were 6148 firearm deaths during the study period. The mean age was 14 years (SD ± 4 years), of which 81% were male and 41% were non-Hispanic White. The most common manners of death were homicide (57%), suicide (36%), and unintentional (7%). Over one-third of firearms were stored unlocked. Homicide deaths had a higher likelihood of occurring outside of the home setting (aOR 3.2, 95% CI 2.4–4.4) compared with unintentional deaths. Suicide deaths had a higher likelihood of occurring in homes with firearms that were stored locked (aOR 4.2, 95% CI 2.1–8.9) compared with unintentional deaths. Each manner of firearm death presents a unique set of psychosocial circumstances and challenges for preventive strategies. Unsafe firearm storage practices remain a central theme in contributing to the increased risk of youth firearm deaths.

## 1. Introduction

Firearm injury affecting youth is a largely preventable public health crisis in the United States. It represents the leading cause of death in children with the highest case-fatality rate among pediatric trauma-related injuries [1]. Firearm injuries in children result in over 20,000 emergency department (ED) visits and 7000 pediatric hospitalizations annually, leading to significant morbidity, disability, and premature mortality, as well as substantial healthcare system burden and cost [2,3].

The US is an outlier when compared to other high-income countries when comparing firearm deaths. Among World Health Organization data for high-income countries, 96.7% of all children aged 0–4 years who died by firearm in 2015 were from the US [4]. This stems from variations in firearm legislation across states regarding ownership and carriage of firearms [5,6]. In both adult and pediatric-based studies, states with more firearm laws had lower rates of firearm-related injuries and fatalities [6,7,8]. In one study, there was a 6-fold difference in magnitude between the highest and lowest state-level mortality rates [6]. Child access prevention (CAP) laws vary by state, holding the caregiver responsible for actions a child takes or potentially takes with a firearm [9]. CAP laws have demonstrated effectiveness in reducing unintentional and suicide firearm deaths in children, with negligence laws having the most significant reduction [8,10].

The American Academy of Pediatrics has stated that the most effective means to prevent any type of firearm injury among children is to reduce the presence of guns in homes and communities [11]. Despite this, over 30% of households with children in the United States have firearms [12,13,14]. Among homes with firearms, up to 43% have at least one unlocked firearm, and more than 10% store at least one firearm unlocked and with ammunition [12,13]. Given this level of exposure, evidence suggests that physician-led education of families, in addition to the provision of safe storage devices, can increase firearm safety in homes. In one systematic review, several office-based counseling initiatives, paired with the provision of safe storage devices, were effective at improving safe firearm storage behaviors in households [15].

Information is limited regarding risk factors for firearm deaths in children, especially information related to individual, familial, and psychosocial factors [16,17,18,19,20]. Further research on how these factors relate to homicide, suicide, and unintentional firearm deaths may help identify injury patterns that guide preventive efforts. There are limited data on the link between adverse childhood experiences (ACEs) and how they impact the risk of firearm injury [21]. The linkage between disproportionate health outcomes and ACEs has traditionally focused on factors such as child maltreatment, residential instability, parental divorce, and/or history of incarcerated family members [22]. More recent studies have examined the involvement of Child Protective Services (CPS), the juvenile justice system, and community violence as ACEs [21,23,24]. The identification of exposure to gun violence as an ACE has garnered interest; however, a lack of federal funding in this area has slowed progress [21]. Less is known about how ACEs and factors related to firearm exposure differ across firearm deaths classified as unintentional, suicide, or homicide [16]. The recognition of injury patterns by mechanisms underlying firearm death, such as the role of firearm ownership, storage, and psychosocial factors, help to inform prevention and policy [25,26].

The purpose of this study is to compare circumstances contributing to unintentional, suicidal, and homicidal firearm deaths in children. In particular, we sought to understand how risk factors related to unintentional deaths differ from intentional homicide and suicide deaths. Unintentional firearm deaths are less common than suicide or homicide deaths but are often viewed as preventable [14,19,27]. Therefore, the unintentional group was selected as our comparison group due to its lack of planning/intent compared with the other groups. A further understanding of these factors will help clinicians, researchers, and public health entities direct preventive interventions, such as risk screening and/or targeted distribution of firearm safety/storage devices, geared toward households with firearms and children.

## 2. Materials and Methods

### 2.1. Study Design

We performed a secondary analysis of previously collected data from the National Fatality Review Case Reporting System (NFR-CRS). We obtained de-identified data on all firearm-related deaths among children aged 0–18 years that occurred during the 10-year period of 2007–2016. We included all firearm-related deaths where the manner of death was homicide, suicide, or unintentional. Deaths with manner documented as natural, undetermined, pending, or unknown were excluded. Suicide deaths under the age of 7 years were excluded due to concern for misclassification, the rarity of suicide, and the lack of routine universal screening for mental health risk factors in this age group [28]. We also excluded deaths in those less than 1 day old and in utero deaths. Data from 34 states met the inclusion criteria and were available in the de-identified research dataset.

There are over 2600 variables available in the NFR-CRS. The variables for this study were first selected based on relevance to our study population using a comprehensive literature review [17,26]. We then excluded variables that were deemed to be redundant or had a large number of missing values to the extent that the data were uninformative. Our final list of variables focused on three broad categories that included demographics, psychosocial factors, and firearm characteristics. Demographic factors included age, sex, race, ethnicity, and urban/rural setting, as well as the specific location of the incident and the decedent’s relationship with the firearm owner. Psychosocial factors in the decedent’s personal and family history included mental health/cognitive disability, alcohol/substance abuse, criminal behavior, caregiver receiving social assistance, history of maltreatment, open CPS case at the time of death, and history of weapon offenses (in decedent or family member). Firearm characteristics included firearm type, license status, safety features, and storage. Detailed descriptions of all the variables are listed in the NFR-CRS Data Dictionary [29].

### 2.2. Description of the Dataset

Data for this analysis were obtained from the NFR-CRS, a web-based system that records and standardizes information from Child Death Review (CDR) team meetings [30]. The CDR is an expansive review of factors contributing to deaths in children in an effort to improve and promote preventive measures [31]. In contrast to psychological autopsies, data from CDR meetings contain more descriptive information on the circumstances of a child’s death, including demographic factors and psychosocial factors. For deaths due to firearms, the data also contain details pertaining to the type of firearm, ownership, storage, and safety features, as well as the circumstances of the incident. The National Center for Fatality Review and Prevention (NCFRP) is a data resource center that offers training and technical support to CDR teams and maintains the NFR-CRS. Forty-seven states currently use the NFR-CRS; three states choose not to participate for various reasons, including state laws or regulations that prohibit participation.

### 2.3. Analysis

Descriptive analyses based on counts (proportion), mean (SD), or median (IQR) were first performed on the decedents’ risk factors grouped by the manner of death. These were followed by two separate logistic regression analyses conducted to identify predictors that differed between each intentional group (homicide, suicide) and the unintentional comparison group, expressing the strength of association as odds ratios (ORs) and 95% confidence intervals (95% CIs). Covariates had varying degrees of missingness due to unknown responses or underreporting from data submitted by individual CDR teams. A multiple imputation (MI) analysis was performed to estimate missing values based on covariate profiles under the missing at-random assumption within each group using the MI package in Stata 16 [32]. Due to the interrelationship among several variables of interest, we first tested for collinearity. Based on the MI analysis results, the following variables were excluded from our model due to multicollinearity: mental health/cognitive health history, alcohol/substance abuse history, history of criminal behavior, history of weapon offenses, licensed firearm, firearm stored with ammunition, and firearm stored loaded. The following variables that had over 70% missing data were also excluded from the logistic regression analyses: firearm safety features and firearm locked (homicide vs. unintentional), and caregiver receiving social services (homicide vs. unintentional, suicide vs. unintentional). Using the remaining predictors, we implemented multiple logistic modeling controlling for year of death, age, race/ethnicity, gender, and setting to identify characteristics that differed between homicide/suicide deaths and unintentional deaths based on adjusted odds ratios with 95% CIs. Due to the hypothesis-generating nature of this study, we identified associations that may not meet the strict definition of statistical significance, such as including 1 in the extreme of the 95% CI around an estimated odds ratio.

## 3. Results

During our 10-year study period, there were a total of 6148 firearm deaths in children, and 6024 met the inclusion criteria for this study (Figure 1). Demographic, psychosocial, and firearm characteristics by manner of death are shown in Table 1. The mean age for all decedents was 14 years (SD ± 4 years). The decedents were predominantly non-Hispanic (NH) White (41%) and male (81%). The majority of deaths occurred in an urban setting (69%). Rural deaths predominated in the suicide (33%) and unintentional (40%) groups compared with homicides (10%). A history of mental health and/or cognitive disability was most prevalent in suicide deaths (33%). A history of criminal behavior was more common in the homicide group (29%). Deaths were predominantly in the home setting, with the majority of these deaths occurring in the decedent’s home (45%). This was most prevalent in suicide (74%) and unintentional (53%) deaths.

Handguns (64%) were the most common firearm used, followed by shotguns (11%) and hunting rifles (6%). Large proportions (57–77%) of licensing and safety variables were missing, but still, a higher proportion of unintentional deaths (21%) involved unlicensed firearms compared to homicide (14%) and suicide (12%) deaths. When reported, most firearms were without safety features, stored unlocked in all groups (35%), with the highest prevalence among the unintentional group (70%). Firearms stored loaded and with ammunition were especially common in unintentional deaths followed by suicides.

Demographic characteristics in homicide and suicide deaths vs. unintentional deaths are shown in Table 2. Unintentional deaths were more likely to occur in decedents aged 14 years and under compared with homicide and suicide deaths. Relative to the unintentional group, homicide deaths were far more likely to occur in an urban setting (OR 6.6, 95% CI 5.2–8.3) and among children of NH-Black (OR 5.9, 95% CI 4.6–7.5), Hispanic racial (OR 5.0, 95% CI 3.6–6.9), and ethnic groups (OR 2.3, 95% CI 1.4–3.8). Suicide deaths were more common in NH-White children compared with NH-Black (OR 0.2, 95% CI 0.2–0.3) and Hispanic children (OR 0.7, 95% CI 0.5–0.9) in the unintentional group, with no sizable difference in setting.

In subsequent analyses, results adjusted for demographic differences were presented alongside unadjusted comparisons using multiple imputed and raw data. In evaluating the relationships, the adjusted results based on imputed data were given priority. Table 3 and Table 4 compare the adjusted and unadjusted psychosocial and firearm characteristics in the homicide and suicide vs. unintentional deaths, respectively, using raw and imputed data.

In Table 3, homicide deaths were more likely to occur outside of the home setting (aOR 3.2, 95% CI 2.4–4.4) using firearms not owned by the decedent or family members (aOR 0.2, 95% CI 0.1–0.3). Homicide decedents were also more likely to have a history of child maltreatment (aOR 1.3, 95% CI 1.0–1.8). They were less likely to have died by hunting rifles (aOR 0.5, 95% 0.2–1.0) than unintentional deaths, with no significant differences in other firearm types.

In Table 4, suicide deaths relative to unintentional deaths involved firearms that were far more likely to belong to the decedent or a family member (aOR 6.8, 95% CI 4.2–10.8) and stored locked (aOR 4.2, 95% CI 2.1–8.9). Suicide was also much less likely to occur outside of the decedent’s home compared with a friend’s home (aOR 0.1, 95% CI 0.1–0.1), relative’s home (aOR 0.3, 95% CI 0.2–0.6), or “other” home (aOR 0.4, 95% CI 0.3–0.6) in unintentional deaths. Documented history of child maltreatment and an open CPS case at the time of death were similar across both groups. There were no significant differences in the type of firearm used or firearm safety features present between suicide and unintentional groups.

## 4. Discussion

The NFR-CRS results affirm our understanding of the individual and familial-level factors contributing to firearm deaths in children, especially that certain psychosocial, and environmental circumstances predispose to each manner of firearm death. Homicide deaths are more likely to occur in an urban setting, at public locations, and among racial/ethnic minorities. In contrast, firearms in homes play a central role in unintentional and suicide deaths, especially firearms stored unlocked in unintentional firearm deaths. These results underscore the enormous risk of having firearms in households with children.

Although the NFR-CRS offers only a snapshot of deaths in the study period, it contains fairly representative data when compared to the Web-based Injury Statistics and Query Reporting System (WISQARS). The WISQARS had 20 045 firearm deaths (vs. 6024 in the NFR-CRS) in all 50 states during the study period, with similar compositions of homicide (62% vs. 57% in NFR-CRS), suicide (32% vs. 36%), and unintentional deaths (5% vs. 7%). Similarly, homicides were most common in NH-Black children (60% vs. 56% in NFR-CRS), while suicides occurred most often in NH-White children (78% vs. 75%) [3].

Consistent with prior studies, unintentional firearm deaths in children are more common in younger White males and occur primarily in the home setting [14,19,27,33,34]. Close to one-third of unintentional decedents were under the age of 5 years and half of the unintentional deaths occurred in the decedent’s home with their own or a family member’s firearm [14,33,34]. Furthermore, over two-thirds of firearms were stored unlocked, and approximately half were stored with ammunition or loaded. These findings emphasize the importance of firearm safety including safe storage and firearm safety features, which were in place in only 12% of unintentional deaths after accounting for missing data. This is particularly critical for US states with higher firearm availability, more lenient gun laws, and increased firearm-related mortality and injury severity in children [7,8,27,34]. More widespread adoption of negligence CAP laws may have the potential for reducing unintentional firearm injuries [8,10].

Our results are consistent with prior studies reporting that the majority of homicidal deaths occur in an urban setting among Black males with an average age of 14 years [20,33,35]. Up to one-quarter of homicide decedents were victims of maltreatment. A history of substance abuse and criminal behavior, especially weapon offenses, were more common in victims of homicide. Further research is needed on the effectiveness of community-based violence interventions in youth, such as the role of social media in reducing firearm outcomes and programs aimed at reducing firearm carriage [17,26]. These have demonstrated promise in the past and have become a large focus in recent grant funding for firearm research [26,36,37,38].

Suicide victims were far more likely to have died from firearms that belonged to the decedent or a family member. Approximately one-quarter of decedents in suicide deaths had preceding mental health concerns. This further elucidates the importance of safe firearm storage to be universally applied, as only a subset of suicide decedents had reported mental health diagnoses at their CDR team assessment. Lethal means screening and counseling are important tasks in the healthcare setting, but knowledge gaps still remain regarding the best approach for lethal means counseling and efficacy [17,26].

A comprehensive preventive strategy is essential for reducing the medical and public health burden of firearm injuries in the US. Many preventive strategies have focused on universal screening and education-based interventions in healthcare settings. The effectiveness of this “one size fits all” approach has been debated, especially because of its limited impact on reducing firearm injuries [19,26,39,40,41]. An alternative approach places emphasis on developing specific screening tools tailored to an individual’s and community’s risk profile and vulnerability; for instance, screening for unsafe firearm storage practices to define strategies, such as the provision of lock boxes and/or trigger locks, to mitigate the higher risk of unintentional firearm injuries and deaths in households with children aged 10 years and under in states with lenient gun laws. The cost of a trigger lock can be as low as USD 10. Both community- and clinic-based preventive interventions demonstrated improved firearm safe storage practices with the distribution of free safety devices (e.g., lock boxes, trigger locks) [15]. Further research is required on determining the efficacy of each type of firearm safety device and the best approach to motivate households to use them [15,17]. More widespread recognition of exposure to gun violence as an ACE would also lead to a higher level of resources for research and federal funding, as ACEs are a lead funding priority area for the Centers for Disease Control and Prevention (CDC) [21]. Although this has improved in recent years, with approximately USD 2,939,768 in total funding allocated to firearm violence topics for 2022, this remains a rate-limiting step in more far-reaching research priorities [42].

There are several potential limitations that may impact our study. Not all states participate in the NFR-CRS, not all participating states review all child deaths, and some states do not make their data available for research purposes. Consequently, mortality rates cannot be calculated using these omitted data. Some NFR-CRS variables have a high proportion of missing data. To account for this, we estimated missing values using multiple imputation. The lack of substantial differences between imputed and non-imputed data is consistent with the missing at-random assumption and supports the adequacy of the data for drawing unbiased effect estimates. Key variables with very large proportions of missing data were not imputed and were excluded from further analyses. Data from 34 states met the inclusion criteria for this study, and they may not be fully representative of all firearm fatalities among children in the US. Despite these limitations, the breadth of information and level of detail available in the NFR-CRS for all manners of death is not available in any other mortality data. This provides a unique opportunity to identify potential risk and protective factors important for future study of firearm fatalities among youth and may inform development of prevention strategies.

## 5. Conclusions

The NFR-CRS highlights the complexities affecting firearm deaths in children. Each manner of firearm death is accompanied by unique circumstances that need to be understood and addressed with preventive efforts. Homicide deaths were more likely among racial ethnic/racial minorities in urban environments, highlighting the importance of further research on the effectiveness of community-based violence interventions. After adjusting for age, race/ethnicity, year of death, and urban/rural setting, the risk factor profiles in young firearm decedents were found to differ between intentional (homicide and suicide) and unintentional outcomes in ways that potentially help identify persons at risk. Firearm homicides are more likely than unintentional deaths to involve documented victims of maltreatment that may not be supported by open CPS cases. Suicide and unintentional deaths tend to have similar risk factor profiles; both were more likely to occur in the home environment predominantly with firearms that originate in the home. Unlocked firearms were far more likely among unintentional deaths, emphasizing the need for increased primary prevention and motivation for caregivers to improve home firearm safety. The absence of a risk difference between the mechanism underlying firearm death does not rule out the importance of a potential risk factor between victims and comparable non-victims. In spite of many challenges, it is important to take every opportunity to identify risk factors, evaluate the impact of targeted preventive strategies, and allocate resources to the most vulnerable households and communities in order to mitigate the burden of this public health crisis.

## Figures and Tables

**Figure 1 children-10-01359-f001:**
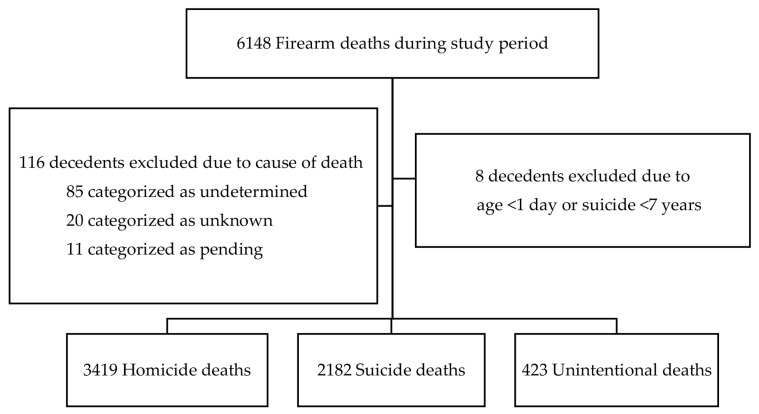
Firearm Decedents Flow Diagram.

**Table 1 children-10-01359-t001:** Demographic, Psychosocial, and Firearm Characteristics of Decedents by Method of Firearm Death.

	Total (n = 6024)	Homicide (n = 3419)	Suicide (n = 2182)	Unintentional (n = 423)
Demographics				
Age (years)				
Mean (±SD)	14 (4)	14 (5)	16 (2)	10 (6)
Median (IQR)	16 (14–17)	16 (14–17)	16 (15–17)	11 (4–15)
Total by Age Group				
0–4 years	412 (7%)	284 (8%)	0	128 (30%)
5–9 years	320 (5%)	257 (8%)	Omitted ^1^	61 (15%)
10–14 years	1151 (19%)	504 (15%)	543 (25%)	106 (25%)
15–18 years	4141 (69%)	2374 (69%)	1639 (75%)	128 (30%)
Sex				
Male	4899 (81%)	2710 (79%)	1834 (84%)	355 (84%)
Female	1104 (18%)	699 (20%)	339 (16%)	66 (16%)
Missing	21 (<1%)	10 (<1%)	9 (<1%)	2 (<1%)
Race/Ethnicity				
Black, Non-Hispanic	2216 (37%)	1922 (56%)	176 (8%)	118 (28%)
Hispanic	953 (16%)	674 (20%)	230 (11%)	49 (12%)
Other	252 (4%)	123 (4%)	97 (4%)	21 (5%)
White, Non-Hispanic	2504 (41%)	638 (19%)	1636 (75%)	230 (54%)
Missing	99 (2%)	51 (1%)	43 (2%)	5 (1%)
Setting				
Urban	4162 (69%)	2729 (80%)	1228 (56%)	205 (48%)
Rural	1225 (20%)	340 (10%)	717 (33%)	168 (40%)
Missing	637 (11%)	350 (10%)	237 (11%)	50 (12%)
Psychosocial Characteristics				
Mental Health/Cognitive Disability History				
Yes	1187 (20%)	452 (13%)	712 (33%)	23 (5%)
No	1907 (32%)	1024 (30%)	693 (32%)	190 (45%)
Missing	2930 (48%)	1943 (57%)	777 (35%)	210 (50%)
Alcohol/Substance Abuse History				
Yes	1175 (19%)	711 (21%)	436 (20%)	28 (7%)
No	2141 (36%)	1099 (32%)	800 (37%)	242 (57%)
Missing	2708 (45%)	1609 (47%)	946 (43%)	153 (36%)
History of Criminal Behavior ^2^				
Yes	1291 (22%)	977 (29%)	284 (13%)	30 (7%)
No	2852 (47%)	1412 (41%)	1152 (53%)	288 (68%)
Missing	1881 (31%)	1030 (30%)	746 (34%)	105 (25%)
Caregiver Receiving Social Assistance ^3^				
Yes	667 (11%)	496 (15%)	116 (5%)	55 (13%)
No	1048 (17%)	411 (12%)	552 (25%)	85 (20%)
Missing	4309 (72%)	2512 (73%)	1514 (70%)	283 (67%)
History of Maltreatment ^4^				
Yes	1213 (20%)	811 (24%)	342 (16%)	60 (14%)
No	2402 (40%)	1243 (36%)	958 (44%)	201 (48%)
Missing	2409 (40%)	1365 (40%)	882 (40%)	162 (38%)
Open Child Protective Services Case at Time of Death				
Yes	265 (4%)	188 (6%)	62 (3%)	15 (4%)
No	4195 (70%)	2306 (67%)	1587 (73%)	302 (71%)
Missing	1564 (26%)	925 (27%)	533 (24%)	106 (25%)
History of weapon offenses ^5^				
Yes	523 (9%)	431 (12%)	72 (3%)	20 (5%)
No	2356 (39%)	846 (25%)	1258 (58%)	252 (59%)
Missing	3145 (52%)	2142 (63%)	852 (39%)	151 (36%)
Location of Incident				
Decedent’s Home	2724 (45%)	892 (26%)	1609 (74%)	223 (53%)
Relative’s Home	319 (5%)	159 (5%)	109 (5%)	51 (12%)
Friend’s Home	533 (9%)	385 (11%)	81 (4%)	67 (16%)
Other ^6^	2339 (39%)	1903 (56%)	363 (16%)	73 (17%)
Missing	109 (2%)	80 (2%)	20 (1%)	9 (2%)
Relationship to Firearm Owner				
Self or Family Member	2276 (38%)	580 (17%)	1440 (66%)	256 (60%)
Other	1210 (20%)	1045 (31%)	95 (4%)	70 (17%)
Missing	2538 (42%)	1794 (52%)	647 (30%)	97 (23%)
Firearm Characteristics				
Firearm Type				
Handgun	3829 (64%)	2230 (65%)	1317 (61%)	282 (67%)
Shotgun	634 (11%)	204 (6%)	371 (17%)	59 (14%)
Hunting Rifle	383 (6%)	76 (2%)	263 (12%)	44 (10%)
Assault Rifle	128 (2%)	96 (3%)	25 (1%)	7 (2%)
Other ^7^	70 (1%)	34 (1%)	30 (1%)	6 (1%)
Missing	980 (16%)	779 (23%)	176 (8%)	25 (6%)
Licensed Firearm				
Yes	916 (15%)	305 (9%)	517 (24%)	94 (22%)
No	846 (14%)	493 (14%)	265 (12%)	88 (21%)
Missing	4262 (71%)	2621 (77%)	1400 (64%)	241 (57%)
Safety features				
Yes	369 (6%)	128 (4%)	189 (9%)	52 (12%)
No	1340 (22%)	696 (20%)	517 (24%)	127 (30%)
Missing	4315 (72%)	2595 (76%)	1476 (67%)	244 (58%)
Locked storage				
Yes	305 (5%)	36 (1%)	261 (12%)	8 (2%)
No	2126 (35%)	877 (26%)	951 (44%)	298 (70%)
Missing	3593 (60%)	2506 (73%)	970 (44%)	117 (28%)
Store with Ammunition				
Yes	1253 (21%)	397 (12%)	661 (30%)	195 (46%)
No	248 (4%)	52 (1%)	165 (8%)	31 (7%)
Missing	4523 (75%)	2970 (87%)	1356 (62%)	197 (47%)
Stored Loaded				
Yes	999 (17%)	407 (12%)	362 (17%)	230 (54%)
No	379 (6%)	55 (2%)	295 (13%)	29 (7%)
Missing	4646 (77%)	2957 (86%)	1525 (70%)	164 (39%)

^1^—Counts and percentages suppressed to protect decedent identities. Numbers were added to the 10–14-year age group. ^2^—History of delinquency or involvement in criminal activity including assaults, robberies, drugs, and juvenile detention. ^3^—One or both caregivers receiving social assistance including Medicaid, WIC, TANF, or food stamps. ^4^—History of documented physical, sexual, or emotional abuse. ^5^—Weapon offenses either in the decedent or family member. ^6^—Most common “other” locations include public roads, sidewalks, parking areas, public parks, and recreational areas. ^7^—Includes assault rifles, pellet guns, and air rifles.

**Table 2 children-10-01359-t002:** Unadjusted OR for Demographic Variables.

Decedent Characteristic	Homicide vs. Unintentional	Suicide vs. Unintentional
	Unadjusted OR (95% CI)	Unadjusted OR (95% CI)
Age		
15–18 years	Reference	Reference
0–4 years	**0.1 (0.1–0.2)**	Omitted ^1^
5–9 years	**0.2 (0.2–0.3)**	Omitted ^1^
10–14 years	**0.3 (0.1–0.3)**	**0.4 (0.3–0.5)**
Sex		
Male	--------	--------
Female	**1.4 (1.1–1.8)**	1.0 (0.7–1.3)
Race/Ethnicity		
White, Non-Hispanic	Reference	Reference
Black, Non-Hispanic	**5.9 (4.6–7.5)**	**0.2 (0.2–0.3)**
Hispanic	**5.0 (3.6–6.9)**	**0.7 (0.5–0.9)**
Other	**2.3 (1.4–3.8)**	0.7 (0.4–1.1)
Setting		
Rural	Reference	Reference
Urban	**6.6 (5.2–8.3)**	**1.4 (1.1–1.8)**
Missing	**3.5 (2.4–4.9)**	1.1 (0.8–1.6)

^1^—Suicide deaths under the age of 7 years were excluded from this analysis.

**Table 3 children-10-01359-t003:** Homicide vs. Unintentional Multivariate Logistic Regression Model.

	Multiple Imputation		Non-Imputed	
Decedent	Unadjusted OR	Adjusted OR ^1^	Unadjusted OR	Adjusted OR ^1^
Characteristic	(95% CI)	(95% CI)	(95% CI)	(95% CI)
Documented Maltreatment				
No	-------	-------	-------	-------
Yes	**1.7 (1.3–2.3)**	**1.3 (1.0–1.8)**	**2.2 (1.6–3.0)**	**1.6 (1.2–2.2)**
Open Child Protective Services Case at the Time of Death				
No	--------	--------	--------	--------
Yes	1.2 (0.7–2.0)	1.2 (0.7–2.1)	1.6 (0.9–2.8)	1.6 (0.9–3.0)
Incident Location				
Child’s Home	Reference	Reference	Reference	Reference
Friend’s Home	**1.4 (1.1–1.9)**	0.9 (0.6–1.2)	**1.4 (1.1–1.9)**	0.8 (0.6–1.2)
Relative’s Home	0.8 (0.6–1.2)	0.8 (0.6–1.2)	0.8 (0.6–1.1)	0.8 (0.5–1.2)
Other	**6.5 (5.0–8.6)**	**3.2 (2.4–4.4)**	**6.0 (4.6–7.9)**	**3.1 (2.3–4.2)**
Firearm Type				
Other	Reference	Reference	Reference	Reference
Handgun	0.9 (0.5–1.6)	0.9 (0.5–1.6)	0.8 (0.4–1.4)	0.8 (0.4–1.5)
Hunting Rifle	**0.2 (0.1–0.4)**	**0.5 (0.2–1.0)**	**0.2 (0.1–0.3)**	**0.4 (0.2–0.9)**
Shotgun	**0.4 (0.2–0.7)**	0.8 (0.4–1.6)	**0.3 (0.2–0.7)**	0.7 (0.4–1.5)
Relationship to Firearm Owner				
Other	--------	--------	--------	--------
Self or Family	**0.1 (0.1–0.1)**	**0.2 (0.1–0.3)**	**0.2 (0.1–0.2)**	**0.3 (0.2–0.4)**

^1^—Adjusted for year of death, age, race/ethnicity, gender, and setting.

**Table 4 children-10-01359-t004:** Suicide vs. Unintentional Multivariate Logistic Regression Model.

	Multiple Imputation		Non-Imputed	
Decedent	Unadjusted OR	Adjusted OR ^1^	Unadjusted OR	Adjusted OR ^1^
Characteristic	(95% CI)	(95% CI)	(95% CI)	(95% CI)
Documented Maltreatment				
No	--------	--------	--------	--------
Yes	1.0 (0.7–1.3)	0.9 (0.6–1.3)	1.2 (0.9–1.6)	1.0 (0.6–1.5)
Open Child Protective Services Case at the Time of Death				
No	--------	--------	--------	--------
Yes	0.7 (0.4–1.2)	1.3 (0.6–3.1)	0.8 (0.4–1.4)	1.7 (0.7–4.2)
Incident Location				
Child’s Home	Reference	Reference	Reference	Reference
Friend’s Home	**0.2 (0.1–0.2)**	**0.1 (0.1–0.1)**	**0.2 (0.1–0.2)**	**0.1 (0.1–0.1)**
Relative’s Home	**0.3 (0.2–0.4)**	**0.3 (0.2–0.6)**	**0.3 (0.2–0.4)**	**0.2 (0.2–0.6)**
Other	**0.7 (0.5–0.9)**	**0.4 (0.3–0.6)**	**0.6 (0.5–0.9)**	**0.4 (0.2–0.5)**
Firearm Type				
Other	Reference	Reference	Reference	Reference
Handgun	1.3 (0.7–2.3)	**2.2 (1.0–4.6)**	1.1 (0.6–2.0)	**2.0 (1.0–4.3)**
Hunting Rifle	1.6 (0.8–3.0)	2.2 (0.9–5.1)	1.4 (0.7–2.8)	2.1 (0.9–4.8)
Shotgun	1.6 (0.9–3.1)	2.1 (0.9–4.6)	1.5 (0.8–2.9)	2.0 (0.9–4.4)
Relationship to Firearm Owner				
Other	--------	--------	--------	--------
Self or Family	**3.9 (2.8–5.6)**	**6.8 (4.2–10.8)**	**4.1 (3.0–5.8)**	**8.2 (5.2–12.8)**
Firearm Safety Features				
Yes	--------	--------	--------	--------
No	1.1 (0.7–1.6)	1.0 (0.6–1.8)	1.1 (0.8–1.6)	1.1 (0.7–1.9)
Firearm Locked				
No	--------	--------	--------	--------
Yes	**6.7 (3.2–13.6)**	**4.2 (2.1–8.9)**	10.2 (0.5–20.1)	**5.6 (2.5–12.3)**

^1^—Adjusted for year of death, age, race/ethnicity, gender, and setting.

## Data Availability

The data that support the findings of this study are available upon request from the corresponding author.

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
