# Peer review of "Firearm Deaths among Youth in the United States, 2007–2016"

_children, 2023, doi:10.3390/children10081359_

Round 1
Reviewer 1 Report
Dear authors,
I am enclosing the report of the article. I found it very interesting but I would like to suggest some improvements.
Regards

Reviewer 2 Report
Thank you for the opportunity to review this manuscript. Firearm injuries among children is a serious problem, especially in the USA. The gap that this study sought to address was risk factors to injuries caused by firearms. This information is relevant for the formulation of policies and therefore it makes the study very relevant. I have concerns and suggestions that may strengthen this manuscript.
1. In lines 20-21, you state, ‘There were 6024 firearm deaths during the study period.’ This is not correct. This is the number that met your inclusion criteria. Please correct the statement.
2. In Table 2 you state that suicide under 7 years was excluded from the analysis, yet you provide unadjusted OR for the 5-9 years age group, which includes the under 7 years old. Please correct this as it will confuse the reader.
3. In lines 170-175, you state, ‘Unintentional deaths were more likely to occur in decedents aged 0-14 years compared with homicide and suicide deaths. Relative to the unintentional group, homicide deaths were far more likely to occur in an urban setting, and among children of NH-Black and Hispanic racial and ethnic groups. Suicide deaths were more common in NH-White children compared to the unintentional group, with no sizable difference in setting.’ Add the unadjusted OR in brackets for each finding.
4. In lines 188-189, you state, ‘by firearms not owned by the decedent or family members (aOR 0.2, 95% CI 0.1-0.3).’ However, this information is not found in Table 3 so remove it from here.
5. In lines 189-190, you state, ‘The three most common settings were public roads, sidewalks, and parking areas.’ This information is also not found in Table 3 so remove it from here.
6. In line 190, you state, ‘Homicide decedents were also more likely to have a history of child maltreatment (aOR 1.3, 95% CI 1.0-1.8).’ The 95% CI includes 1.0, meaning there is no statistical difference. I would not say ‘more likely’ here.
7. In line 191-192, you state, ‘They were less likely to have died by hunting rifles (aOR 0.5, 95% 0.2-1.0) than unintentional deaths…’ The 95% CI includes 1.0, meaning there is no statistical difference. I would not say ‘less likely’ here.
8. In line 197, you state, ‘Suicide was also much less likely to 196 occur outside of the decedent’s home.’ Include the aOR and the 95% CI.
9. In tables, significant OR or aOR and their 95% CI should be bolded for the readers to easily identify them.
Round 2
Reviewer 2 Report
All my comments have been addressed.